# Teaching Multiple Concepts to a Forgetful Learner

**Anette Hunziker**[†]    **Yuxin Chen**[¶]    **Oisin Mac Aodha**[§]    **Manuel Gomez Rodriguez**[*]
**Andreas Krause**[‡]    **Pietro Perona**[⋆]    **Yisong Yue**[⋆]    **Adish Singla**[*]

[†]University of Zurich, `anette.hunziker@gmail.com`,
[¶]University of Chicago, `chenyuxin@uchicago.edu`,
[§]University of Edinburgh, `oisin.macaodha@ed.ac.uk`,
[‡]ETH Zurich, `krausea@ethz.ch`,
[⋆]Caltech, `{perona, yyue}@caltech.edu`,
[*]MPI-SWS, `{manuelgr, adishs}@mpi-sws.org`

## Abstract

How can we help a forgetful learner learn multiple concepts within a limited time frame? While there have been extensive studies in designing optimal schedules for teaching a *single* concept given a learner's memory model, existing approaches for teaching *multiple* concepts are typically based on heuristic scheduling techniques without theoretical guarantees. In this paper, we look at the problem from the perspective of discrete optimization and introduce a novel algorithmic framework for teaching multiple concepts with strong performance guarantees. Our framework is both generic, allowing the design of teaching schedules for different memory models, and also interactive, allowing the teacher to adapt the schedule to the underlying forgetting mechanisms of the learner. Furthermore, for a well-known memory model, we are able to identify a regime of model parameters where our framework is guaranteed to achieve high performance. We perform extensive evaluations using simulations along with real user studies in two concrete applications: (i) an educational app for online vocabulary teaching; and (ii) an app for teaching novices how to recognize animal species from images. Our results demonstrate the effectiveness of our algorithm compared to popular heuristic approaches.

## 1   Introduction

In many real-world educational applications, human learners often intend to learn more than one concept. For example, in a language learning scenario, a learner aims to memorize many vocabulary words from a foreign language. In citizen science projects such as eBird [34] and iNaturalist [38], the goal of a learner is to recognize multiple animal species from a given geographic region. As the number of concepts increases, the learning problem can become very challenging due to the learner's limited memory and propensity to forget. It has been well established in the psychology literature that in the context of *human learning*, the knowledge of a learner decays rapidly without reconsolidation [7]. Somewhat analogously, in the sequential *machine learning* setting, modern machine learning methods, such as artificial neural networks, can be drastically disrupted when presented with new information from different domains, which leads to catastrophic interference and forgetting [19, 14]. Therefore, to retain long-term memory (for both human and machine learners), it is crucial to devise teaching strategies that adapt to the underlying forgetting mechanisms of the learner.

Teaching forgetful learners requires repetition. Properly scheduled repetitions and reconsolidations of previous knowledge have proven effective for a wide variety of real-world learning tasks, including piano [30], surgery [39, 33], video games [29], and vocabulary learning [4], among others. For many of the above applications, it has been shown that by carefully designing the scheduling policy, one can

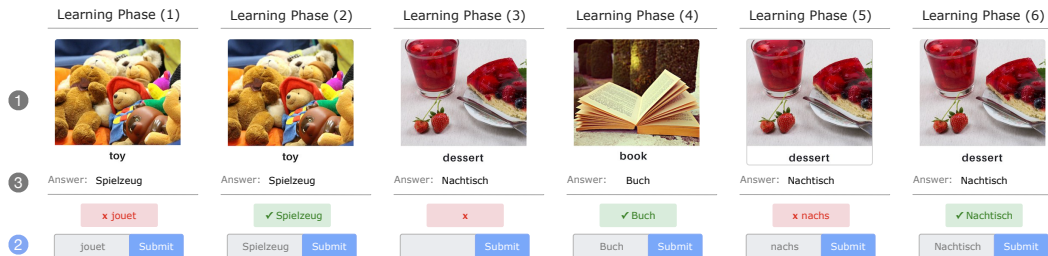

Figure 1: Illustration of our adaptive teaching framework applied to German vocabulary learning, shown here for six time steps in the learning phase. Each time step proceeds in three stages: (1) the system displays a flashcard with an image and its English description, (2) the learner inputs the German translation, and (3) the system provides feedback in the form of the correct answer if the input is incorrect.

achieve substantial gains over simple heuristics (such as spaced repetition at fixed time intervals, or a simple round robin schedule) [3]. Unfortunately, while there have been extensive (theoretical) results in teaching a single concept using spaced repetition algorithms, existing approaches for teaching multiple concepts are typically based on heuristics without theoretical guarantees.

In this paper, we explore the following research question: *Given limited time, can we help a forgetful learner efficiently learn multiple concepts in a principled manner?* More concretely, we consider an adaptive setting where at each time step, the teacher needs to pick a concept from a finite set based on the learner's previous responses, and the process iterates until the learner's time budget is exhausted. Given a memory model of the learner, what is an optimal teaching curriculum? How should this sequence be adapted based on the learner's performance history?

## 1.1 Overview of our approach

For a high-level overview of our approach, consider the example in Fig. 1, which illustrates one of our applications on German vocabulary learning [2]. Here, our goal is to teach the learner three German words in six time steps. One trivial approach could be to show the flashcards in a round robin fashion. However, the round robin sequence is deterministic and thus not capable of adapting to the learner's performance. In contrast, our algorithm outputs an adaptive teaching sequence based on the learner's performance.

Our algorithm is based on a novel formulation of the adaptive teaching problem. In §2, we propose a novel discrete optimization problem, where we seek to maximize a natural surrogate objective function that characterizes the learner's expected performance throughout the teaching session. Note that constructing the optimal teaching policy boils down to solving a stochastic sequence optimization problem, which is NP-hard in general. In §3, we introduce our greedy algorithm, and derive performance guarantees based on two intuitive data-dependent properties. While it can be challenging to compute these performance bounds, we show that for certain learner memory models, these bounds can be estimated efficiently. Furthermore, we identify parameter settings of the memory models where the greedy algorithm is guaranteed to achieve high performance. Finally, we demonstrate that our algorithm achieves significant improvements over baselines for both simulated learners (cf. §4) and human learners (cf. §5).

## 2 The Teaching Model

We now formalize the problem addressed in this paper.

## 2.1 Problem setup

Suppose that the teacher aims to teach the learner $n$ concepts in a finite time horizon $T$. We highlight the notion of a concept via two concrete examples: (i) when teaching the vocabulary of a foreign language, each concept corresponds to a word, and (ii) when teaching to recognize different animal species, each concept corresponds to an animal name. We consider flashcard-based teaching, where each concept is associated with a flashcard (cf. Fig. 1).

We study the following interactive teaching protocol: At time step $t$, the teacher picks a concept from the set $\{1, \ldots, n\}$ and presents its corresponding flashcard to the learner without revealing its correct answer. The learner then tries to recall the concept. Let us use $y_t \in \{0, 1\}$ to denote the learner's recall at time step $t$. Here, $y_t = 1$ means that the learner successfully recalls the concept (e.g., the learner correctly recognizes the animal species), and $y_t = 0$ otherwise. After the learner makes an attempt, the teacher observes the outcome $y_t$ and reveals the correct answer.

## 2.2 Learner's memory model

Let us use $(\sigma, y)$ to denote any sequence of concepts and observations. In particular, we use $\sigma_{1:t}$ to denote the sequence of concepts picked by the teacher up to time $t$. Similarly, we use $y_{1:t}$ to denote the sequence of observations up to time $t$. Given the history $(\sigma_{1:t}, y_{1:t})$, we are interested in modeling the learner's probability to recall concept $i$ at a future time $\tau \in [t + 1, T]$. In general, the learner's probability to recall concept $i$ could depend on the history of teaching concept $i$ or related concepts.[1] Formally, we capture the learner's recall probability for concept $i$ by a memory model $g_i(\tau, (\sigma_{1:t}, y_{1:t}))$ that depends on the entire history $(\sigma, y)$. In §3.2, we study an instance of the learner model captured by exponential forgetting curve (see Eq. (9)).

## 2.3 The teaching objective

There are several objectives of interest to the teacher, for instance, maximizing the learner's performance in recalling all concepts measured at the end of the teaching session. However, given that the learning phase might stretch over a long time duration for language learning, another natural objective is to measure learner's performance across the entire teaching session. For any given sequence of concepts and observations $(\sigma_{1:T}, y_{1:T})$ of length $T$, we consider the following objective:

$$f(\sigma_{1:T}, y_{1:T}) = \frac{1}{nT} \sum_{i=1}^{n} \sum_{\tau=1}^{T} g_i(\tau + 1, (\sigma_{1:\tau}, y_{1:\tau})). \tag{1}$$

Here, $g_i(\cdot)$ denotes the recall probability of concept $i$ at $\tau + 1$, given the history up to time step $\tau$. Concretely, for a given concept $i \in [n]$, our objective function can be interpreted as the (discrete) area under the learner's recall curve for concept $i$ across the teaching session.

The teacher's teaching strategy can be represented as a *policy* $\pi : (\sigma, y) \to \{1, \ldots, n\}$, which maps any history (i.e., sequence of concepts selected $\sigma$ and observations $y$) to the next concept to be taught. For a given policy $\pi$, we use $(\sigma_{1:T}^{\pi}, y_{1:T}^{\pi})$ to denote a random trajectory from the policy until time $T$. The average utility of a policy $\pi$ is defined as:

$$F(\pi) = \mathbb{E}_{\sigma^{\pi}, y^{\pi}}[f(\sigma_{1:T}^{\pi}, y_{1:T}^{\pi})]. \tag{2}$$

Given the learner's memory model for each concept $i$ and the time horizon $T$, we seek the optimal teaching policy that achieves the maximal average utility:

$$\pi^* \in \max_{\pi} F(\pi). \tag{3}$$

It can be shown that finding the optimal solution for Eq. (3) is NP-hard (proof is provided in the supplemental materials).

**Theorem 1.** *Problem (3) is NP-hard, even when the objective function does not depend on the learner's responses.*

## 3 Teaching Algorithm and Analysis

We now present a simple, greedy approach for constructing teaching policies. To measure the teaching progress at time $t < T$, we introduce the following generalization of objective defined in Eq. (1):

$$f(\sigma_{1:t}, y_{1:t}) = \frac{1}{nT} \sum_{i=1}^{n} \sum_{\tau=1}^{T} g_i \left( \tau + 1, \left( \sigma_{1:\min(\tau,t)}, y_{1:\min(\tau,t)} \right) \right). \tag{4}$$

Note that this is equivalent to extending $(\sigma_{1:t}, y_{1:t})$ to length $T$ by filling in the remaining sequence from $t+1$ to $T$ with empty concepts and observations. Given the history $(\sigma_{1:t-1}, y_{1:t-1})$, we define the conditional marginal gain of teaching a concept $i$ at time $t$ as:

$$\Delta\left(i \mid \sigma_{1:t-1}, y_{1:t-1}\right) = \mathbb{E}_{y_t}\left[f(\sigma_{1:t-1} \oplus i, y_{1:t-1} \oplus y_t) - f(\sigma_{1:t-1}, y_{1:t-1})\right], \tag{5}$$

where $\oplus$ denotes the concatenation operation, and the expectation is taken over the randomness of learner's recall $y_t$, conditioned on the history $(\sigma_{1:t-1}, y_{1:t-1})$. The greedy algorithm, as described in Algorithm 1, iteratively selects the concept that maximizes this conditional marginal gain.

---
**Algorithm 1** Adaptive Teaching Algorithm
---
Sequence $\sigma \leftarrow \emptyset$; observation history $y \leftarrow \emptyset$
**for** $t = \{1, \ldots, T\}$ **do**
    Select $i_t \leftarrow \arg\max_i \Delta\left(i \mid \sigma, y\right)$
    Show $i_t$ to the learner; Observe $y_t$
    Update $\sigma \leftarrow \sigma \oplus i_t, y \leftarrow y \oplus y_t$

---

### 3.1 Theoretical guarantees

We now present a general theoretical framework for analyzing the performance of the adaptive teaching algorithm (Algorithm 1). Our bound depends on two natural properties of the objective function $f$, both related to a notion of *diminishing returns* of a sequence function. Intuitively, the following two properties reflect how much a greedy choice can affect the optimality of the solution.

**Definition 1** (Online stepwise submodular coefficient). *Consider policy $\pi$ for time $T$. The online submodular coefficient of function $f$ with respect to policy $\pi$ at step $t$ is defined as*

$$\gamma_t^\pi = \min_{\sigma_{1:t}^\pi, y_{1:t}^\pi} \gamma(\sigma_{1:t}^\pi, y_{1:t}^\pi) \tag{6}$$

*where $\gamma(\sigma, y) = \min_{i,(\sigma',y'):|\sigma|+|\sigma'|<T} \frac{\Delta(i|\sigma,y)}{\Delta(i|\sigma\oplus\sigma',y\oplus y')}$ denotes the minimal ratio between the gain of any concept $i$ given the current history $(\sigma, y)$ and the gain of $i$ in any future steps.*

**Definition 2** (Online stepwise backward curvature). *Consider policy $\pi$ for time $T$. The online backward curvature of function $f$ with respect to policy $\pi$ at step $t$ is defined as*

$$\omega_t^\pi = \max_{\sigma_{1:t}^\pi, y_{1:t}^\pi} \omega(\sigma_{1:t}^\pi, y_{1:t}^\pi) \tag{7}$$

*where $\omega(\sigma, y) = \max_{\sigma^{\pi'}, y^{\pi'}} \left[1 - \frac{f(\sigma \oplus \sigma^{\pi'}, y \oplus y^{\pi'}) - f(\sigma^{\pi'}, y^{\pi'})}{f(\sigma,y) - f(\emptyset)}\right]$ denotes the normalized maximal second-order difference when considering the current history $(\sigma, y)$.*

Here, $\gamma(\sigma, y)$ and $\omega(\sigma, y)$ generalizes the notion of *string submodularity* and *total backward curvature* for sequence functions [43] to the stochastic setting. Intuitively, $\gamma(\sigma, y)$ measures the degree of diminishing returns of a sequence function in terms of the *ratio* between the conditional marginal gains. If $\forall(\sigma, y), \gamma(\sigma, y) = 1$, then the conditional marginal gain of adding any concept to any subsequent observation history is non-decreasing. In contrast, $\omega(\sigma, y)$ measures the degree of diminishing returns in terms of the *difference* between the marginal gains. As our first main theoretical result, we provide a data-dependent bound on the average utility of the greedy policy against the optimal policy.

**Theorem 2.** *Let $\pi^g$ be the online greedy policy induced by Algorithm 1, and $F$ be the objective function as defined in Eq. (2). Then for all policies $\pi^*$,*

$$F\left(\pi^g\right) \geq F\left(\pi^*\right) \sum_{t=1}^{T} \left(\frac{\gamma_{T-t}^g}{T} \prod_{\tau=0}^{t-1} \left(1 - \frac{\omega_\tau^g \gamma_\tau^g}{T}\right)\right), \tag{8}$$

*where $\gamma_t^g$ and $\omega_t^g$ denote the online stepwise submodular coefficient and online stepwise backward curvature of $f$ with respect to the policy $\pi^g$ at time step $t$.*

The summand on the R.H.S. of Eq. (8) is in fact a lower bound on the expected one-step gain of the greedy policy. We can further relax the bound by considering the worst-case online stepwise submodularity ratio and curvature across all time steps, given by the following corollary.

**Corollary 3.** *Let* $\gamma^g = \min_t \gamma_t^g$ *and* $\omega^g = \max_t \omega_t^g$. *For all* $\pi^*$, $F(\pi^g) \geq \frac{1}{\omega^g}\left(1 - e^{-\gamma^g \omega^g}\right) F(\pi^*)$.

Note that Corollary 3 generalizes the string submodular optimization result from [43] to the stochastic setting. In particular, for the special case where $\gamma^g = \omega^g = 1$ and $f(\sigma_{1:t}, y_{1:t})$ is independent of $y_{1:t}$, Corollary 3 reduces to $f(\sigma^g, \cdot) \geq \left(1 - e^{-1}\right) f(\sigma^*, \cdot)$ where $\sigma^g, \sigma^*$ denote the sequences selected by the greedy and the optimal algorithm. However, constructing the bounds in Theorem 2 and Corollary 3 requires us to compute $\gamma_t^g, \omega_t^g$, which is as expensive as computing $F(\pi^*)$. In the following subsection, we investigate a specific learning setting, and provide a polynomial time approximation algorithm for computing the theoretical lower bound in Theorem 2.

## 3.2 Analysis for HLR memory model

Here, we consider the setting where the learner's memory for each concept $i \in [n]$ is captured by an independent HLR memory model. Concepts being independent means that the memory model $g_i(\tau, (\sigma_{1:t}, y_{1:t}))$ for concept $i$ only depends on the history when flashcards for concept $i$ was shown.[2]

More specifically, we consider the case of an HLR memory model with the following exponential forgetting curve [28]:

$$g_i(\tau, (\sigma_{1:t}, y_{1:t})) = 2^{-\frac{\tau - \ell_i}{h_i}}, \tag{9}$$

where $\ell_i$ is the last time concept $i$ was taught, and $h_i = 2^{\langle \theta_i, n_i \rangle}$ denotes the half life of the learner's recall probability of concept $i$. Here, $\theta_i = (a_i, b_i, c_i)$ parameterizes the learner's retention rate, and $n_i = (n_+^i, n_-^i, 1)$, where $n_+^i := |\{\tau' \in [t] : \sigma_{\tau'} = i \ \wedge \ y_{\tau'} = 1\}|$ and $n_-^i := |\{\tau' \in [t] : \sigma_{\tau'} = i \ \wedge \ y_{\tau'} = 0\}|$ denote the number of correct and incorrect recalls of concept $i$ in $(\sigma_{1:t}, y_{1:t})$, respectively. Intuitively, $a_i$ scales $n_+^i$, $b_i$ scales $n_-^i$, and $c_i$ is an offset that can be considered as scaling time.

We would like to bound the performance of Algorithm 1. While computing $\gamma_t^g, \omega_t^g$ is intractable in general, we show that one can efficiently approximate $\gamma_t^g, \omega_t^g$ for the HLR model with $a_i = b_i$.

**Theorem 4.** *Assume that the learner is characterized by the HLR model (Eq. (9)) where* $\forall i, \ a_i = b_i$. *We can compute empirical bounds on* $\gamma_t, \omega_t$ *in polynomial time.*

Theorem 4 shows that it is feasible to compute explicit lower bounds on the utility of Algorithm 1 against the maximal achievable utility. The following theorem shows that for certain types of learners, the algorithm is guaranteed to achieve a high utility.

**Theorem 5.** *Consider the task of teaching $n$ concepts where each concept is following an independent HLR memory model sharing the same parameters, i.e.,* $\forall \ i, \theta_i = (a, a, 0)$. *A sufficient condition for the algorithm to achieve $1 - \epsilon$ utility is* $a \geq \max\left\{\log T, \log(3n), \log\left(\frac{2n^2}{\epsilon T}\right)\right\}$, *where the parameter $a$ essentially captures the learner's memory strength.*

Note that Theorem 5 provides a sufficient condition for our algorithm to achieve a high utility. One interesting open question is to establish an upper bound for the greedy (or the optimal) algorithm under particular model configurations, e.g., to provide a necessary condition for achieving a certain target utility under the HLR model.

## 4 Simulations

In this section, we experimentally evaluate our algorithm by simulating learner responses based on a known memory model. This allows us to inspect the behavior of our algorithm and compare it with several baseline algorithms in a controlled setting.

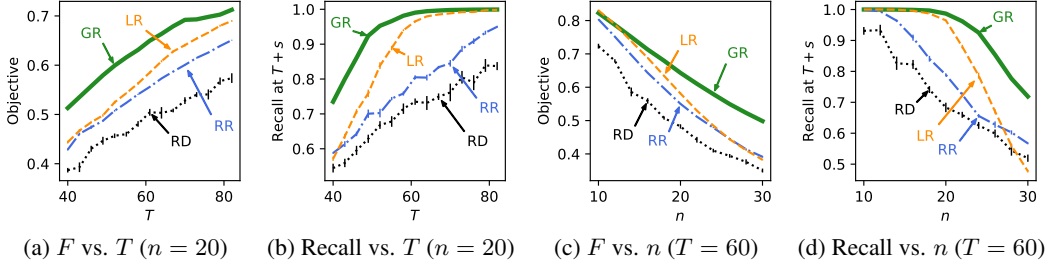

| (a) $F$ vs. $T$ ($n = 20$) | (b) Recall vs. $T$ ($n = 20$) | (c) $F$ vs. $n$ ($T = 60$) | (d) Recall vs. $n$ ($T = 60$) |

Figure 2: Simulation results comparing our algorithm (GR) and three baseline algorithms (RD, RR, and LR). Two performance metrics are considered: (i) the objective value in Eq. (4) and (ii) recall at the end of the teaching session denoted at 'Recall at $T + s$" with $s = 10$.

## 4.1 Experimental setup

**Dataset** We simulated concepts of two different types: "easy" and "difficult". The learner's memory for each concept is captured by an independent HLR model. Concepts of the same type share the same parameter configurations. Specifically, for "easy" concepts the parameters are $\theta_1 = (a_1 = 10, b_1 = 5, c_1 = 0)$, and for "difficult" concepts the parameters are $\theta_2 = (a_2 = 3, b_2 = 1.5, c_2 = 0)$, with the following interpretation in terms of recall probabilities. For "easy" concepts, the recall probability of a concept $i$ drops to $g_i\left(\tau = 2, (\sigma_1 = i, y_1 = 1)\right) = 2^{-1/(2^{a_1+c_1})} = 0.99$ and $g_i\left(\tau = 2, (\sigma_1 = i, y_1 = 0)\right) = 2^{-1/(2^{b_1+c_1})} = 0.98$ in the immediate next step after showing concept $i$. For "difficult" concepts these probabilities are $(0.92, 0.78)$.

**Evaluation metric** We consider two different criteria when assessing the performance. Our first evaluation metric is the objective value as defined in Eq. (4), which measures the learner's average cumulative recall probability *across* the entire teaching session. The second evaluation metric is the learner's average recall probability at the *end* of the teaching session. We call this second objective "Recall at $T + s$", where $s > 0$ denotes how far in the future we choose to evaluate the learner's recall.

**Baselines** To demonstrate the performance of our adaptive greedy policy (referred to as GR), we consider three baseline algorithms. The first baseline, denoted by RD, is a random teacher that presents a random concept at each time step. The second baseline, denoted by RR, is a round robin teaching policy that picks concepts according to a fixed round robin schedule, i.e., iterating through concepts at each time step. Our third baseline is a variant of the teaching strategy employed by [28], which can be considered as a generalization of the popular Leitner and Pimsleur systems [16, 25]. At each time step, the teacher chooses to display the concept with the lowest recall probability according to the HLR memory model of the learner. We refer to this algorithm as LR.

## 4.2 Simulation results

We first evaluate the performance as a function of the teaching horizon $T$. In Fig. 2a and Fig. 2b, we plot the objective value and average recall at $T + s$ for all algorithms over 10 random trials, where we set $s = 10$, $n = 20$ with half easy and half difficult concepts, and vary $T \in [40, 80]$. As we can see from both plots, GR consistently outperforms baselines in all scenarios. The gap between the performances of GR and the baselines is more significant for smaller $T$. As we increase the time budget, the performance of all algorithms improves—this behavior is expected, as it corresponds to the scenario where all concepts get a fair chance of repetition with abundant time budget. In Fig. 2c and Fig. 2d, we show the performance plot for a fixed teaching horizon of $T = 60$ when we vary the number of concepts $n \in [10, 30]$. Here we observe a similar behavior as before—GR is consistently better; as $n$ increases, the gap between the performances of GR and the baselines becomes more significant. Our results suggest that the advantage of GR is most pronounced for more challenging settings, i.e., when we have a tight time budget (small $T$) or a large number of concepts (large $n$).

| | German | | | | Biodiversity | | | |
|---|---|---|---|---|---|---|---|---|
| | GR | LR | RR | RD | GR | LR | RR | RD |
| avg gain | 0.572 | 0.487 | 0.462 | 0.467 | 0.475 | 0.411 | 0.390 | 0.251 |
| $p$-value | – | 0.0652 | 0.0197 | 0.0151 | – | 0.0017 | <0.0001 | <0.0001 |

| | Biodiversity (common) | | | | Biodiversity (rare) | | | |
|---|---|---|---|---|---|---|---|---|
| | GR | LR | RR | RD | GR | LR | RR | RD |
| avg gain | 0.143 | 0.118 | 0.150 | 0.086 | 0.766 | 0.668 | 0.601 | 0.396 |
| $p$-value | – | 0.3111 | 0.8478 | 0.0047 | – | 0.0001 | <0.0001 | <0.0001 |

Table 1: Summary of the user study results. Here, the performance is measured as the gain in learner's performance from prequiz phase to postquiz phase (see main text for details). We have $n = 15, T = 40$, and ran algorithms with a total of 80 participants for German app and 320 participants for Biodiversity app.

# 5 User Study

We have developed online apps for two concrete real-world applications: (i) German vocabulary teaching [2], and (ii) teaching novices to recognize animal species from images, motivated by citizen science projects for biodiversity monitoring [1]. Next, we briefly introduce the datasets used for these two apps and then present the user study results of teaching human learners.

## 5.1 Experimental setup

**Dataset**   For the German vocabulary teaching app, we collected 100 English-German word pairs in the form of flashcards, each associated with a descriptive image. These word pairs were provided by a language expert (see [8]) and consist of popular vocabulary words taught in an entry-level German language course. For the biodiversity teaching app, we collected images of 50 animal species. To extract a fine-grained signal for our user study, we further categorize the Biodiversity dataset into two difficulty levels, namely "common" and "rare", based on the prevalence of these species. Examples from both datasets are provided in the supplemental materials.

For real-world experiments, we do not know the learner's memory model. While it is possible to fit the HLR model through an extensive pre-study as in [28], we instead simply choose a fixed set of parameters. For the Biodiversity dataset, we set the parameters of each concept based on their difficulty level. Namely, we set $\theta_1 = (10, 5, 0)$ for "common" (i.e., easy) species and $\theta_2 = (3, 1.5, 0)$ for "rare" (i.e., difficult) species, as also used in our simulation. For the German dataset, since the parameters associated with a concept (i.e., vocabulary word) depend heavily on learner's prior knowledge, we chose a more robust set of parameters for each of the concepts given by $\theta = (6, 2, 0)$. We defer the details of our sensitivity study of the HLR parameters to the supplemental materials.

**Online teaching interface**   Our apps provide an online teaching interface where a user (i.e., human learner) can participate in a "teaching session". As in the simulations, here each session corresponds to teaching $n$ concepts (sampled randomly from our dataset) via flashcards over $T$ time steps. We demonstrate the teaching interface and present the detailed design ideas in the supplemental materials.

## 5.2 User study results

**Results for German**   We now present the user study results for our German vocabulary teaching app [2]. We run our candidate algorithms with $n = 15, T = 40$ on a total of 80 participants (i.e., 20 per algorithm) recruited from Amazon Mechanical Turk. Results are shown in Table 1. where we computed the average gain of each algorithm, and performed statistical analysis on the collected results. The first row (*avg gain*) is obtained by treating the performance for each (participant, word) pair as a separate sample (e.g., we get $20 \times 15$ samples per algorithm for the German app). The second row (*p-value*) indicates the statistical significance of the results measured by the $\chi^2$ tests [6] (with contingency tables where rows are algorithms and columns are observed outcomes), when comparing GR with the baselines. Overall, GR achieved higher gains compared to the baselines.

**Results for Biodiversity**  Next, we present the user study results on our Biodiversity teaching app [1]. We recruited a total of 320 participants (i.e., 80 per algorithm). Here, we used different parameters for the learner's memory as described in §5.1; all other conditions (i.e., $n = 15$, $T = 40$, and interface) were kept the same as for the German app. In Table 1, in addition to the overall performance of the algorithms across all concepts, we also provide separate statistics on teaching the "common" and "rare" concepts. Note that, while the performance of GR is close to the baselines when teaching the "common" species (given the high prequiz score due to learner's prior knowledge about these species), GR is significantly more effective in teaching the "rare" species.

**Remarks**  This user study provides a proof-of-concept that the performance of our algorithm GR demonstrated on simulated learners is consistent with the performance observed on human learners. While teaching sessions in our current user study were limited to a span of 25 mins with participants recruited from Mechanical Turk, we expect that the teaching applications we have developed could be adapted to real-life educational scenarios for conducting long-term studies.

# 6  Related Work

**Spaced repetition and memory models**  Numerous studies in neurobiology and psychology have emphasized the importance of the *spacing* effects in human learning. The spacing effect is the observation that spaced repetition (i.e., introducing appropriate time gaps when learning a concept) produces greater improvements in learning compared to massed repetition (i.e., "cramming") [37]. These findings have inspired many computational models of human memory, including the Adaptive Character of Thought–Rational model (ACT-R) [24], the Multiscale Context model (MCM) [22], and the Half-life Regression model (HLR) [28]. In particular, HLR is a trainable spaced repetition model, which can be viewed as a generalization of the popular Leitner [16] and Pimsleur [25] systems. In this paper, we adopt a variant of HLR to model the learner. One of the key characteristics of these memory models is the function used to model the forgetting curve. Power-law and exponential functions are two popular ways of modeling the forgetting curve (for detailed discussion, see [27, 41, 24, 40]).

**Optimal scheduling with spaced repetition models**  [13] and [17] studied the ACT-R model and the MCM model respectively for the optimal review scheduling problem where the goal is to maximize a learner's retention through an intelligent review scheduler. One of the key differences between their setting and ours is that, they consider a fixed curriculum of new concepts to teach, and the scheduler additionally chooses which previous concept(s) to review at each step; whereas our goal is to design a complete teaching curriculum. Even though the problem settings are somewhat different, we would like to note that our theoretical framework can be adapted to their setting.

Recently, [26] presented a queuing model for flashcard learning based on the Leitner system and consider a "mean-recall approximation" heuristic to tractably optimize the review schedule. One limitation is that their approach does not adapt to the learner's performance over time. Furthermore, the authors leave the problem of obtaining guarantees for the original review scheduling problem as a question for future work. [35] considered optimizing learning schedules in continuous time for a single concept, and use control theory to derive optimal scheduling to minimize a penalized recall probability area-under-the-curve loss function. In addition to being discrete time, the key difference of our setting is that we aim to teach multiple concepts.

**Sequence optimization**  Our theoretical framework is inspired by recent results on sequence submodular function maximization [43, 36] and adaptive submodular optimization [10]. In particular, [43] introduced the notion of string submodular functions, which, analogous to the classical notion of submodular set functions [15], enjoy similar performance guarantees for maximization of deterministic sequence functions. Our setting has two key differences in that we focus on the stochastic setting with potentially non-submodular objective functions. In fact, our theoretical framework (in particular Corollary 3) generalizes string submodular function maximization to the adaptive setting.

**Forgetful learners in machine learning**  Here, we highlight the differences with some recent work in the machine learning literature involving forgetful learners. In particular, [44] aimed to teach the learner a binary classifier by sequentially providing training examples, where the learner has an exponential decaying memory of the training examples. In contrast, we study a different problem, where we focus on teaching multiple concepts, and assume that the learner's memory of each concept

decays over time. [14] explored the problem of how to construct a neural network for learning multiple concepts. Instead of designing the optimal training schedule, their goal is to design a good learner that suffers less from the forgetting behavior.

**Machine teaching**   Our work is also closely related to machine/algorithmic teaching literature (e.g., [46, 45, 32, 9]). Most of these works in machine teaching consider a non-adaptive setting where the teacher provides a batch of teaching examples at once without any adaptation. In this paper, we focus primarily on designing interactive teaching algorithms that adaptively select teaching examples for a learner based on their responses. The problem of adaptive teaching has been studied recently (e.g., [12, 42, 11, 5, 18, 31]). However, these works in machine teaching have not considered the phenomena of forgetting. [23, 21] have studied the problem of concept learning and machine teaching when learner has "limited-capacity" in terms of retrieving exemplars in memory during the decision-making process. They model the learner via the Generalized Context Model [20] and investigated the problem of choosing the optimal exemplars for teaching a *classification* task. In our setting, the exemplars for each class are already given (in other words, we have only one exemplar per class), and we aim at optimally teaching the learner to *memorize* the (label of) exemplars.

## 7   Conclusions

We presented an algorithmic framework for teaching multiple concepts to a forgetful learner. We proposed a novel discrete formulation of teaching based on stochastic sequence function optimization, and provided a general theoretical framework for deriving performance bounds. We have implemented teaching apps for two real-world applications. We believe our results have made an important step towards bringing the theoretical understanding of algorithmic teaching closer to real-world applications where the forgetting phenomenon is an intrinsic factor.

**Acknowledgements**

This work was done when Yuxin Chen and Oisin Mac Aodha were at Caltech. This work was supported in part by NSF Award #1645832, Northrop Grumman, Bloomberg, AWS Research Credits, Google as part of the Visipedia project, and a Swiss NSF Early Mobility Postdoctoral Fellowship.

## Footnotes

[1] As an example, for German vocabulary learning, the recall probability for the concept "Apfelsaft" (apple juice) could depend on the flashcards shown for "Apfelsaft" and "Apfel" (apple).

[2]We note that the hardness result of Theorem 1 doesn't directly apply to this setting with independent concepts. Nevertheless, the problem setting is still computationally challenging. If we express the optimal solution using Bellman equations and apply dynamic programming, the number of states will be exponential in the number of concepts $n$ and polynomial w.r.t. time horizon $T$.

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
