[Supplementary Material · neurips19_teaching-forgetful_camera-ready-sup.pdf]

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

# A  List of Appendices

In this section we provide a brief description of the content provided in the appendices of the paper.

- Appendix B provides details of the sensitivity analysis of our model.

- Appendix C provides details of the user study.

- Appendix D provides further performance analysis for the greedy algorithm when teaching an HLR learner.

- Appendix E provides proofs of the theoretical results.

# B  Robustness and Sensitivity Analysis

We conducted a sensitivity study on simulated learners before choosing the HLR parameters for our user study. These detailed results are demonstrated in Fig. 3.

Figure 3: Sensitivity analysis of our teaching algorithm

In this experiment, we consider two groups of concepts: "easy/common" concepts with $\theta = (10, 5, 0)$, and "hard/rare"concepts with $\theta = (3, 1.5, 0)$. Other configurations are kept the same as our user study, with $T = 40$, $n = 15$ and $s = 10$.

We vary the number of "easy" concepts from $\{0, 1, \ldots, 8\}$ (i.e., up to 50% of the concepts being easy), and consider four types of teachers: (i) "easy": $\theta = (10, 5, 0)$ for all concepts; (ii) "hard": $\theta = (3, 1.5, 0)$ for all concepts; (iii) "true": using the true parameters for each concept; (iv) "robust": $\theta = (6, 2, 0)$ for all concepts. We plot the performances of these different teachers measured by the two metrics considered in simulations (i.e., the objective value and future recall). As shown in the figures, the "robust" teacher performs well on both metrics, and hence is used for our user study on the German dataset.

# C  User Study

## C.1  Online teaching interface

We set up a simple and adaptive interface to keep the learners engaged in our user study (see Fig. 4). To establish a setup that accurately reflects our modeling assumptions, we integrate the following design ideas.

An important component of the user evaluation is to understand the learner's bias (or prior knowledge), which we cannot easily assess purely based on the learner's inputs during the learning phase. To resolve this issue, we introduce a prequiz phase (before the learning phase starts) where we test the learner's knowledge of concepts in the session by asking them to provide answers for all $n$ concepts. After the learning phase, the learner will enter a postquiz (i.e., testing) phase. By recording the change in the learner's performance from prequiz to postquiz phase, we can estimate the gain of the teaching session.

In order to align the online teaching session with our discrete-time problem formulation, we impose a minimum and maximum time window for each flashcard presentation during the learning phase. In particular, a participant has a maximum time of 20 seconds to provide input, and has 10 seconds to review the correct answer provided by the teacher.

Figure 4: The teaching interface of our online apps (German and Biodiversity).

Another important aspect is the short-term memory effect. In general, it is non-trivial to carry out large scale user studies that span over weeks/months (even though it better fits the HLR model of the learner). Given the physical constraints of our real-world experiments, we consider shorter teaching sessions of 25 mins in duration, involving the teaching of $n = 15$ concepts for a total number of $T = 40$ time steps. To mitigate the short-term memory effect present in our experiments, we impose an additional constraint for the user study[3], such that the algorithms do not pick the same concept for two consecutive time steps (otherwise, a learner will simply "copy" the answer she sees on the previous screen).

## C.2  User study results

Fig. 5a and Fig. 5b illustrates the distribution of learners' performances. Even though some learners failed to achieve good performance, GR managed to teach a larger fraction of learners to achieve better performance compared to the baselines—this suggests that our algorithm is an effective strategy for teaching vocabulary.

(a) German

(b) Biodiversity

Figure 5: User study results

## C.3 Datasets

In this subsection, we show a few samples from both the German dataset (for the German vocabulary teaching app) in Fig. 6, and the Biodiversity dataset (for the biodiversity teaching app) in Fig. 7.

**German dataset**

Figure 6: Samples from the German dataset

**Biodiversity dataset**

(a) Common: Owl, Cat, Horse, Elephant, Lion, Tiger, Bear

(b) Rare: Angwantibo, Olinguito, Axolotl, Ptarmigan, Patrijshond, Coelacanth, Pyrrhuloxia

Figure 7: Samples from the Biodiversity dataset

# D    Teaching Algorithm and Analysis

## D.1    Analysis for HLR memory model

Figure 8: Performance analysis for the greedy algorithm when teaching an HLR learner with $T = 15$ and $n = 3$. Each colored marker from Fig. 8a–8c represents a different concept, with $\theta_1 = (2.50, 2.50, 1.26)$ for blue, $(\theta_2 = 1.00, 1.00, -1.00)$ for orange, and $\theta_3 = (0.08, 0.08, -0.88)$ for the green concept. Intuitively, concepts with higher $\theta_i$ values are easier to teach.

In Fig. 8, we demonstrate the behavior of three teaching algorithms on a toy problem with $T = 15, n = 3$. Fig. 8a-8c show the learner's forgetting curve (i.e., recall probabilities) and the sequences selected by three algorithms: Greedy (Algorithm 1), Optimal (the optimal solution for Problem (3)), and Round Robin (a fixed round robin teaching schedule for all concepts). Observe that Greedy starts with easy concepts (i.e., concepts with higher memory retention rates), moves on to teaching new concepts when the learner has "enough" retention for the current concept, and repeats previously shown concepts towards the end of the teaching session. This behavior is similar to the optimal teaching sequence, and achieves higher utility in comparison to the fixed round robin scheduling (Fig. 8d).

In Fig. 8e, we see that the marginal gain of the orange item is increasing in the early stages (as opposed to many classical discrete optimization problems that exhibit the diminishing returns property), which makes the analysis of the greedy algorithm non-trivial. In Fig. 8f and Fig. 8g, we show the empirical bounds on $\gamma_t^{\mathrm{g}}, \omega_t^{\mathrm{g}}$, as well as the exact values of $\gamma_t^{\mathrm{g}}, \omega_t^{\mathrm{g}}$ when running the greedy algorithm. Note that our procedure for computing $\gamma_t^{\mathrm{g}}$ actually outputs the *exact* value of $\gamma_t^{\mathrm{g}}$ (a näive approach to computing $\gamma_t^{\mathrm{g}}$ is via extensive enumeration of all possible teaching sequences).

In Fig. 8h, we plug in the empirical bounds on $\gamma_t^{\mathrm{g}}$ and $\omega_t^{\mathrm{g}}$ to Theorem 2 and Corollary 3, and plot the empirical approximation bounds on $F(\pi^{\mathrm{g}})/F(\pi^*)$ as a function of the teaching horizon $T$. For problem instances with a large teaching horizon $T$, it is infeasible to compute the true approximation bound. However, one can still efficiently compute the empirical approximation bound as a useful indicator of the performance of our algorithm.

# E    Proofs

## E.1    Proof of Theorem 1

In this section, we provide the proof of Theorem 1. We first show that any non-negative string submodular function can be represented as a constant factor of the objective function $f$ as defined in Eq. (4). We then prove the NP-hardness of the optimization problem (Problem (3)) by the NP-hardness result of string submodular optimization [43].

*Proof.* Recall from Eq. (4) that

$$f(\sigma_{1:t}, y_{1:t}) = \frac{1}{nT} \sum_{i=1}^{n} \sum_{\tau=1}^{T} g_i \left( \tau + 1, \left( \sigma_{1:\min(\tau,t)}, y_{1:\min(\tau,t)} \right) \right).$$

In the following, we show how one can represent an arbitrary non-negative string submodular function in the form of the RHS of the above equation (i.e., Eq. (4)). Let $\mu : \Sigma \to \mathbb{R}_{\geq 0}$ be a (non-negative) string submodular function, where $\Sigma$ denote the set of possible sequences defined over $n$ items (i.e., concepts). For a fixed budget $T$, the string submodular optimization problem can be stated as follows [43]:

$$\sigma^* = \underset{\sigma \in \Sigma, |\sigma| = T}{\arg\max} \ \mu(\sigma). \tag{10}$$

For any sequence $\sigma \in \Sigma$, define $t_i(\sigma)$ to be the (time) index of item $i$ in the sequence[4]. That is,

$$t_i(\sigma) = \begin{cases} \text{the index of item } i \text{ in } \sigma & \text{if } i \in \sigma \\ 0 & \text{o.w.} \end{cases}$$

For simplicity, we drop the dependency of $\sigma$ in $t_i(\sigma)$ when it is clear from the context. Define

$$g_i\left(\tau + 1, \left(\sigma_{1:\min(\tau,t)}, \cdot\right)\right) = \begin{cases} \frac{n}{c} \cdot \mu(i) & \text{if } t_i = 1 \\ \frac{nT}{c \cdot (T - t_i + 1)} \cdot \mu(i \mid \sigma_{1:t_i-1}) & \text{if } 1 < t_i \leq \min(\tau, t) \\ 0 & \text{if } t_i > \min(\tau, t) \text{ or } t_i = 0 \end{cases} \tag{11}$$

Here, $\mu(i \mid \sigma_{1:t_i-1}) = \mu(\sigma_{1:t_i-1} \oplus i) - \mu(\sigma_{1:t_i-1})$ denotes the marginal gain of item $i$. Since $\mu_i(\cdot)$ is (string) submodular, we set $c = nT \cdot \max_i \mu(i)$ as a normalizing constant so that $g_i\left(\tau + 1, \left(\sigma_{1:\min(\tau,t)}, \cdot\right)\right) \leq 1$.

Substituting $g_i$ on the RHS of Eq.(4) by Eq.(11), we get

$$
\begin{aligned}
f(\sigma_{1:t}, \cdot) &= \frac{1}{nT} \sum_{i=1}^{n} \sum_{\tau=1}^{T} g_i\left(\tau + 1, \left(\sigma_{1:\min(\tau,t)}, \cdot\right)\right) \\
&\overset{(a)}{=} \frac{1}{nT} \sum_{i \in \sigma_{1:t}} \sum_{\tau=1}^{T} g_i\left(\tau + 1, \left(\sigma_{1:\min(\tau,t)}, \cdot\right)\right) \\
&\overset{(b)}{=} \frac{1}{nT} \sum_{i \in \sigma_{1:t}} (T - t_i + 1) \cdot \frac{nT}{c \cdot (T - t_i + 1)} \cdot \mu(i \mid \sigma_{1:t_i-1}) \\
&= \sum_{i \in \sigma_{1:t}} \frac{1}{c} \cdot \mu(i \mid \sigma_{1:t_i-1}) \\
&= \frac{1}{c} \cdot \mu(\sigma_{1:t})
\end{aligned}
$$

Here, step (a) and (b) are by the definition of $g_i$ in Eq. (11). Therefore, for any sequence $\sigma \in \Sigma$, one can represent $\mu(\sigma)$ in terms of $cf(\sigma, \cdot)$. By the NP-hardness result of string submodular optimization [43], we conclude that the general optimization problem (Problem (3)) is NP-hard. $\qquad \square$

## E.2 Proof of Theorem 2 and Corollary 3

### E.2.1 Notations and definitions

For simplicity, we first introduce the notation which will be used in the proof.

Let us use function $\phi(i, t)$ to represent a learner's recall of item $i$ at $t$, where $\phi(i, t) = 1$ indicates that the learner recalls item $i$ correctly at time $t$, and $\phi(i, t) = 0$ otherwise. We call the function $\phi$ a *realization*, and use $\Phi$ to denote a random realization. A realization $\phi$ is consistent with the observation history $(\sigma_{1:t}, y_{1:t})$, if $\phi(\sigma_\tau, \tau) = y_\tau$ for all $\tau \in \{1, \dots, t\}$. We denote such a case by $\phi \sim (\sigma_{1:t}, y_{1:t})$.

We further use $(\sigma^\pi(\phi), y^\pi(\phi))$ to denote the sequence of items and observations obtained by running policy $\pi$ under realization $\phi$. Here, $\sigma^\pi(\phi)$ denotes the sequence of items selected by $\pi$ if the learner is responding according to $\phi$.

Similarly with the conditional marginal gain of an item (Eq. (5)), we define the conditional marginal gain of a sequence of items as follows.

**Definition 3** (Conditional marginal gain of a sequence). *Given observation history $(\sigma_{1:t}, y_{1:t})$, the conditional marginal gain of a sequence of items $\sigma$ is defined as*

$$\Delta\left(\sigma \mid \sigma_{1:t}, y_{1:t}\right) = \mathbb{E}[f(\sigma_{1:t} \oplus \sigma, y_{1:t} \oplus y) - f(\sigma_{1:t}, y_{1:t}) \mid (\sigma_{1:t}, y_{1:t})]. \tag{12}$$

We also define the conditional marginal gain of a policy.

**Definition 4** (Conditional marginal gain of a policy). *Given observation history $(\sigma_{1:t}, y_{1:t})$, the conditional marginal gain of a policy $\pi$ is defined as*

$$\Delta\left(\pi \mid \sigma_{1:t}, y_{1:t}\right) = \mathbb{E}[f(\sigma_{1:t} \oplus \sigma^\pi(\Phi), y_{1:t} \oplus y^\pi(\Phi)) - f(\sigma_{1:t}, y_{1:t}) \mid \Phi \sim (\sigma_{1:t}, y_{1:t})]. \tag{13}$$

By $\sigma_{1:t} \oplus \sigma^\pi(\Phi)$, we mean concatenating the sequence chosen by $\pi$ under realization $\Phi$ (i.e., $\sigma^\pi(\Phi)$) with some existing history $\sigma_{1:t}$ (note that the first $t$ elements of $\sigma^\pi(\Phi)$ could be completely different from $\sigma_{1:t}$).

### E.2.2 Proof of Theorem 2

To prove Theorem 2, we first establish a lower bound on the one-step gain of the greedy algorithm. The following lemma provides a lower bound of the one-step conditional marginal gain of the greedy policy $\pi^g$ against the conditional marginal gain of any policy (of length $T$).

**Lemma 6.** *Suppose we have selected sequence $\sigma_{1:t}$ and observed $y_{1:t}$. Then, for any policy $\pi$ of length $T$,*

$$\max_i \Delta\left(i \mid \sigma_{1:t}, y_{1:t}\right) \geq \frac{\gamma_t^\pi}{T} \Delta\left(\pi \mid \sigma_{1:t}, y_{1:t}\right) \tag{14}$$

*Proof.* By Definition 4 we know that for all $\pi$ it holds that

$$\Delta\left(\pi \mid \sigma_{1:t}, y_{1:t}\right) = \mathbb{E}[f(\sigma_{1:t} \oplus \sigma_{1:T}^\pi(\Phi), y_{1:t} \oplus y_{1:T}^\pi(\Phi)) - f(\sigma_{1:t}, y_{1:t}) \mid \Phi \sim (\sigma_{1:t}, y_{1:t})]$$

$$\stackrel{(a)}{=} \mathbb{E}\left[\sum_{\tau=1}^{T} (f(\sigma_{1:t} \oplus \sigma_{1:\tau}^\pi(\Phi), y_{1:t} \oplus y_{1:\tau}^\pi(\Phi)) - \right.$$
$$\left. f(\sigma_{1:t} \oplus \sigma_{1:\tau-1}^\pi(\Phi), y_{1:t} \oplus y_{1:\tau-1}^\pi(\Phi))) \mid \Phi \sim (\sigma_{1:t}, y_{1:t})\right]$$

$$= \sum_{\tau=1}^{T} \mathbb{E}\left[f(\sigma_{1:t} \oplus \sigma_{1:\tau}^\pi(\Phi), y_{1:t} \oplus y_{1:\tau}^\pi(\Phi)) - \right.$$
$$\left. f(\sigma_{1:t} \oplus \sigma_{1:\tau-1}^\pi(\Phi), y_{1:t} \oplus y_{1:\tau-1}^\pi(\Phi)) \mid \Phi \sim (\sigma_{1:t}, y_{1:t})\right]$$

$$\stackrel{(b)}{=} \sum_{\tau=1}^{T} \mathbb{E}\left[\mathbb{E}\left[f(\sigma_{1:t} \oplus \sigma_{1:\tau}^\pi(\Phi'), y_{1:t} \oplus y_{1:\tau}^\pi(\Phi')) - \right.\right.$$
$$f(\sigma_{1:t} \oplus \sigma_{1:\tau-1}^\pi(\Phi'), y_{1:t} \oplus y_{1:\tau-1}^\pi(\Phi'))$$
$$\left.\left. \mid \Phi' \sim (\sigma_{1:t} \oplus \sigma_{1:\tau-1}^\pi(\Phi), y_{1:t} \oplus y_{1:\tau-1}^\pi(\Phi))\right] \mid \Phi \sim (\sigma_{1:t}, y_{1:t})\right]$$

$$\stackrel{\text{Eq. (5)}}{=} \sum_{\tau=1}^{T} \mathbb{E}\left[\Delta\left(\sigma_\tau^\pi(\Phi') \mid \Phi' \sim \sigma_{1:t} \oplus \sigma_{1:\tau-1}^\pi(\Phi), y_{1:t} \oplus y_{1:\tau-1}^\pi(\Phi)\right)\right.$$
$$\left. \mid \Phi \sim (\sigma_{1:t}, y_{1:t})\right] \tag{15}$$

Here, step (a) is a telescoping sum, and step (b) is by the law of total expectation.

Further, by the definition of $\gamma_t$ (Definition 1) we know that for all $\pi$ and $\phi$ it holds that

$$\max_i \Delta\left(i \mid \sigma_{1:t}, y_{1:t}\right) \geq \gamma_t^\pi \Delta\left(\sigma_\tau^\pi(\Phi') \mid \Phi' \sim \sigma_{1:t} \oplus \sigma_{1:\tau-1}^\pi(\phi), y_{1:t} \oplus y_{1:\tau-1}^\pi(\phi)\right) \tag{16}$$

Combining Eq. (15) with Eq. (16) to get

$$\Delta\left(\pi \mid \sigma_{1:t}, y_{1:t}\right) \stackrel{\text{Eq. (15)}}{=} \sum_{\tau=1}^{T} \mathbb{E}\left[\Delta\left(\sigma_\tau^\pi(\Phi') \mid \Phi' \sim \sigma_{1:t} \oplus \sigma_{1:\tau-1}^\pi(\Phi), y_{1:t} \oplus y_{1:\tau-1}^\pi(\Phi)\right)\right.$$

$$\left.\phantom{X}\right|\ \Phi \sim (\sigma_{1:t}, y_{1:t})\Big]$$

$$\overset{\text{Eq. (16)}}{\leq} \sum_{\tau=1}^{T} \mathbb{E}\left[\frac{1}{\gamma_t^\pi} \max_i \Delta\left(i \mid \sigma_{1:t}, y_{1:t}\right) \ \Big|\ \Phi \sim (\sigma_{1:t}, y_{1:t})\right]$$

$$= \frac{T}{\gamma_t^\pi} \max_i \Delta\left(i \mid \sigma_{1:t}, y_{1:t}\right)$$

which completes the proof. $\qquad\square$

In the following we provide the proof of Theorem 2.

*Proof of Theorem 2.* By the definition of $\omega_t$ (Definition 2,Eq. (7)) we know that for all $\pi$ it holds that

$$\omega_t \geq 1 - \frac{\mathbb{E}[f(\sigma_{1:t} \oplus \sigma^\pi(\Phi), y_{1:t} \oplus y^\pi(\Phi)) - f(\sigma^\pi(\Phi), y^\pi(\Phi)) \mid \Phi \sim (\sigma_{1:t}, y_{1:t})]}{f(\sigma_{1:t}, y_{1:t})}$$

Therefore, we get

$$\Delta\left(\pi \mid \sigma_{1:t}, y_{1:t}\right) = \mathbb{E}[f(\sigma_{1:t} \oplus \sigma^\pi(\Phi), y_{1:t} \oplus y^\pi(\Phi)) - f(\sigma_{1:t}, y_{1:t}) \mid \Phi \sim (\sigma_{1:t}, y_{1:t})]$$
$$\geq \mathbb{E}[f(\sigma^\pi(\Phi), y^\pi(\Phi)) - \omega_t f(\sigma_{1:t}, y_{1:t}) \mid \Phi \sim (\sigma_{1:t}, y_{1:t})] \qquad (17)$$

Now suppose that we have run greedy policy $\pi^{\text{g}}$ up to time step $t$ and have observed sequence $(\sigma_{1:t}^{\text{g}}, y_{1:t}^{\text{g}})$. Combining Lemma 6 (Eq. (14)) with Eq. (17), we get

$$\max_i \Delta\left(i \mid \sigma_{1:t}^{\text{g}}, y_{1:t}^{\text{g}}\right) = \mathbb{E}\left[f(\sigma_{1:t+1}^{\text{g}}(\Phi), y_{1:t+1}^{\text{g}}(\Phi)) - f(\sigma_{1:t}^{\text{g}}, y_{1:t}^{\text{g}}) \mid \Phi \sim (\sigma_{1:t}^{\text{g}}, y_{1:t}^{\text{g}})\right]$$

$$\geq \frac{\gamma_t}{T} \cdot \mathbb{E}\left[f(\sigma^\pi(\Phi), y^\pi(\Phi)) - \omega_t f(\sigma_{1:t}^{\text{g}}, y_{1:t}^{\text{g}}) \mid \Phi \sim (\sigma_{1:t}^{\text{g}}, y_{1:t}^{\text{g}})\right]$$

which implies

$$\mathbb{E}\left[f(\sigma_{1:t+1}^{\text{g}}(\Phi), y_{1:t+1}^{\text{g}}(\Phi)) \mid \Phi \sim (\sigma_{1:t}^{\text{g}}, y_{1:t}^{\text{g}})\right]$$
$$\geq \frac{\gamma_t}{T} \cdot \mathbb{E}\left[f(\sigma^\pi(\Phi), y^\pi(\Phi)) \mid \Phi \sim (\sigma_{1:t}^{\text{g}}, y_{1:t}^{\text{g}})\right] + \left(1 - \frac{\gamma_t \omega_t}{T}\right) f(\sigma_{1:t}^{\text{g}}, y_{1:t}^{\text{g}}) \qquad (18)$$

Therefore, we get

$$F\left(\pi^{\text{g}}\right) = \mathbb{E}\left[f(\sigma_{1:T}^{\text{g}}(\Phi), y_{1:T}^{\text{g}}(\Phi))\right]$$

$$\overset{(a)}{=} \mathbb{E}\left[\mathbb{E}\left[f(\sigma_{1:T}^{\text{g}}(\Phi), y_{1:T}^{\text{g}}(\Phi)) \mid \Phi \sim (\sigma_{1:T-1}^{\text{g}}(\Phi'), y_{1:T-1}^{\text{g}}(\Phi'))\right]\right]$$

$$\overset{\text{Eq. (18)}}{\geq} \mathbb{E}\left[\frac{\gamma_{T-1}}{T} \cdot \mathbb{E}\left[f(\sigma^\pi(\Phi), y^\pi(\Phi)) \mid \Phi \sim (\sigma_{1:T-1}^{\text{g}}(\Phi'), y_{1:T-1}^{\text{g}}(\Phi'))\right]\right] +$$

$$\mathbb{E}\left[\left(1 - \frac{\gamma_{T-1} \omega_{T-1}}{T}\right) f(\sigma_{1:T-1}^{\text{g}}(\Phi'), y_{1:T-1}^{\text{g}}(\Phi'))\right]$$

$$\overset{(b)}{=} \frac{\gamma_{T-1}}{T} \cdot \mathbb{E}[f(\sigma^\pi(\Phi), y^\pi(\Phi))] + \left(1 - \frac{\gamma_{T-1} \omega_{T-1}}{T}\right) \cdot \mathbb{E}\left[f(\sigma_{1:T-1}^{\text{g}}(\Phi'), y_{1:T-1}^{\text{g}}(\Phi'))\right]$$

$$= \frac{\gamma_{T-1}}{T} \cdot F\left(\pi\right) + \left(1 - \frac{\gamma_{T-1} \omega_{T-1}}{T}\right) \cdot \mathbb{E}\left[f(\sigma_{1:T-1}^{\text{g}}(\Phi), y_{1:T-1}^{\text{g}}(\Phi))\right] \qquad (19)$$

where step (a) and step (b) are by the law of total expectation. Recursively applying Eq. (19) gives us

$$F\left(\pi^{\text{g}}\right) \geq \frac{\gamma_{T-1}}{T} \cdot F\left(\pi\right) + \left(1 - \frac{\gamma_{T-1} \omega_{T-1}}{T}\right) \cdot \mathbb{E}\left[f(\sigma_{1:T-1}^{\text{g}}(\Phi), y_{1:T-1}^{\text{g}}(\Phi))\right]$$

$$\geq \left(\frac{\gamma_{T-1}}{T} + \left(1 - \frac{\gamma_{T-1} \omega_{T-1}}{T}\right) \frac{\gamma_{T-2}}{T}\right) F\left(\pi\right) +$$

$$\left(1 - \frac{\gamma_{T-1} \omega_{T-1}}{T}\right) \left(1 - \frac{\gamma_{T-2} \omega_{T-2}}{T}\right) \mathbb{E}\left[f(\sigma_{1:T-2}^{\text{g}}(\Phi), y_{1:T-2}^{\text{g}}(\Phi))\right]$$

$$\geq \dots$$

$$\geq F\left(\pi\right) \sum_{t=1}^{T-1} \frac{\gamma_{T-t}}{T} \prod_{\tau=1}^{t-1} \left(1 - \frac{\gamma_\tau \omega_\tau}{T}\right)$$

which completes the proof. $\qquad\square$

### E.2.3 Proof of Corollary 3

*Proof of Corollary 3.* Since $\gamma^{\mathrm{g}} = \min_t \gamma_t$ and $\omega^{\mathrm{g}} = \max_t \omega_t$, by Theorem 2 we obtain

$$F\left(\pi^{\mathrm{g}}\right) \geq F\left(\pi\right) \sum_{t=1}^{T} \frac{\gamma_{T-t}}{T} \prod_{\tau=0}^{t-1} \left(1 - \frac{\gamma_\tau \omega_\tau}{T}\right)$$

$$\geq F\left(\pi\right) \frac{\gamma^{\mathrm{g}}}{T} \sum_{t=1}^{T} \left(1 - \frac{\gamma^{\mathrm{g}} \omega^{\mathrm{g}}}{T}\right)^t$$

$$= F\left(\pi\right) \frac{1}{\omega^{\mathrm{g}}} \left(1 - \left(1 - \frac{\gamma^{\mathrm{g}} \omega^{\mathrm{g}}}{T}\right)^T\right)$$

which completes the proof. $\qquad\square$

### E.3 Proof of Theorem 4

In this section, we provide the proof for Theorem 4. In particular, we divide the proof into two parts. In §E.3.1, we propose a polynomial time algorithm which outputs a lower bound on $\gamma_t^g$; in §E.3.2, we provide an upper bound on $\omega_t^{\mathrm{g}}$ which can be computed in linear time.

### E.3.1 Empirical lower bound on $\gamma_t$ for the case $a = b$

Let us use count $(\sigma, i)$ to denote the function that returns the number of times item $i$ appears in sequence $\sigma$. We first show the following lemma.

**Lemma 7.** *Fix $s \leq t$. For any $\sigma' \in \{\sigma : |\sigma| = t, \text{count}\,(\sigma, i) = s\}$, we have*

$$\Delta\left(i \mid \sigma_{t,1:s}^i, \cdot\right) \geq \Delta\left(i \mid \sigma', \cdot\right)$$

*where $\sigma_{t,1:s}^i := \underbrace{i \oplus i \oplus \cdots \oplus i}_{s \text{ times}} \oplus \underbrace{\_ \oplus \_ \oplus \cdots \oplus \_}_{t - s \text{ times}}$ denotes the sequence of items of length $t$, where the first $s$ items are item $i$ and the remaining $t - s$ items are empty.*

*Proof.* By definition of the marginal gain (Eq. (5))

$$\Delta\left(i \mid \sigma, y\right) = \mathbb{E}[f(\sigma_{1:t} \oplus i, y_{1:t} \oplus \Phi(i, t+1)) - f(\sigma_{1:t}, y_{1:t}) \mid \Phi \sim (\sigma_{1:t}, y_{1:t})]$$

For the case $a = b$, the objective function $f$ is independent of the observed outcomes of the learner's recall. That is,

$$\Delta\left(i \mid \sigma_{1:t}, \cdot\right) = f(\sigma_{1:t} \oplus i, \cdot) - f(\sigma_{1:t}, \cdot)$$

$$= \frac{1}{nT} \sum_{i=1}^{n} \sum_{\tau=1}^{T} \{g_i\left(\tau + 1, \sigma_{1:t} \oplus i, \cdot\right) - g_i\left(\tau + 1, \sigma_{1:t}, \cdot\right)\}$$

$$= \frac{1}{nT} \sum_{i=1}^{n} \sum_{\tau=t+1}^{T} \{g_i\left(\tau + 1, \sigma_{1:t} \oplus i, \cdot\right) - g_i\left(\tau + 1, \sigma_{1:t}, \cdot\right)\}$$

Denote $\Sigma_{t,s}^i = \{\sigma : |\sigma| = t, \text{count}\,(\sigma, i) = s\}$. For any $\sigma, \sigma' \in \Sigma_{t,s}^i$, we know that

$$\sum_{\tau=t+1}^{T} g_i\left(\tau + 1, \sigma_{1:t} \oplus i, \cdot\right) = \sum_{\tau=t+1}^{T} g_i\left(\tau + 1, \sigma'_{1:t} \oplus i, \cdot\right)$$

Therefore,

$$\max_{\sigma_{1:t} \in \Sigma_{t,s}^i} \Delta\left(i \mid \sigma_{1:t}, \cdot\right) = \frac{1}{nT} \sum_{i=1}^{n} \sum_{\tau=t+1}^{T} \left\{g_i\left(\tau + 1, \sigma_{1:t} \oplus i, \cdot\right) - \min_{\sigma_{1:t} \in \Sigma_{t,s}^i} g_i\left(\tau + 1, \sigma_{1:t}, \cdot\right)\right\}$$

$$\overset{(a)}{=} \frac{1}{nT} \sum_{i=1}^{n} \sum_{\tau=t+1}^{T} \left\{g_i\left(\tau + 1, \sigma_{1:t} \oplus i, \cdot\right) - g_i\left(\tau + 1, \sigma_{t,1:s}^i, \cdot\right)\right\}$$

Here, step (a) is due to the fact that the learner's recall of an item is monotonously decreasing (therefore showing item $i$ earlier leads to lower recall in the future). Therefore, it completes the proof. $\qquad\square$

**Algorithm 2** Computing the empirical lower bound on the greedy online stepwise submodular coefficient

---

**Require:** $\sigma_{1:t}; y_{1:t}$
  **for** $i = \{1, \ldots, n\}$ **do**
    $\texttt{CurrentGain}_i \leftarrow \Delta\left(i \mid \sigma_{1:t}, y_{1:t}\right)$
    **for** $\tau = \{1, \ldots, T - t\}$ **do**
      **for** $s \in \{1, \ldots, \tau\}$ **do**
        $\sigma' \leftarrow \underbrace{i \oplus i \oplus \cdots \oplus i}_{s \text{ times}} \oplus \underbrace{\_ \oplus \_ \oplus \cdots \oplus \_}_{\tau - s \text{ times}}$          ▷ Only consider insertions in the front
        $v_{\tau,s} \leftarrow \Delta\left(i \mid \sigma_{1:t} \oplus \sigma', \cdot\right)$          ▷ Gain of item $i$ at $t + \tau$, with $s$ insertions
    $\texttt{FutureGain}_i \leftarrow \max_{\tau,s} v_{\tau,s}$          ▷ Maximal gain of item $i$ at future time steps
  $\gamma_t \leftarrow \min_i \frac{\texttt{CurrentGain}_i}{\texttt{FutureGain}_i}$          ▷ Choosing the minimal ratio among all items
  **return** $\gamma_t$

---

An approximation algorithm for $\gamma_t$ is provided in Algorithm 2.

### E.3.2 Empirical upper bound on $\omega_t$ for the case $a = b$

In this section, we derive an upper bound on $\omega_t$ which can be computed in polynomial time.

By definition of the online greedy stepwise backward curvature $\omega_t$, we know

$$\omega_t := \omega(\sigma^{\text{g}}_{1:t}, y^{\text{g}}_{1:t}) = \max_{\sigma^\pi, y^\pi} \left\{ 1 - \frac{f(\sigma^{\text{g}}_{1:t} \oplus \sigma^\pi, y^{\text{g}}_{1:t} \oplus y^\pi) - f(\sigma^\pi, y^\pi)}{f(\sigma^{\text{g}}_{1:t}, y^{\text{g}}_{1:t})} \right\}$$

For the case $a = b$, the objective function $f$ is independent of the observed outcomes of the learner's recall (i.e., $f$ is a deterministic function of the input teaching sequence). Therefore,

$$\omega_t = \max_\pi \left\{ 1 - \frac{f(\sigma^{\text{g}}_{1:t} \oplus \sigma^\pi, \cdot) - f(\sigma^\pi, \cdot)}{f(\sigma^{\text{g}}_{1:t}, \cdot)} \right\}$$

$$= 1 + \max_\pi \left\{ \frac{f(\sigma^\pi, \cdot) - f(\sigma^{\text{g}}_{1:t} \oplus \sigma^\pi, \cdot)}{f(\sigma^{\text{g}}_{1:t}, \cdot)} \right\}$$

For simplicity let us use $\sigma^{\text{g}+\pi} := \sigma^{\text{g}}_{1:t} \oplus \sigma^\pi$ to denote the concatenated sequence, and w.l.o.g, assume that $\pi$ represent the one which maximizes the RHS of the above equation (i.e., $\pi$ is the optimal policy). Substituting the objective function $f$ in the above equation with its definition (Eq. (4)), we get

$$\omega_t = 1 + \frac{1}{nT} \frac{1}{f(\sigma^{\text{g}}_{1:t}, \cdot)} \sum_{i=1}^n \sum_{\tau=1}^T \left\{ g_i\left(\tau + 1, \sigma^\pi_{1:\tau}, \cdot\right) - g_i\left(\tau + 1, \sigma^{\text{g}+\pi}_{1:\tau}, \cdot\right) \right\}$$

$$= 1 + \frac{1}{nT} \frac{1}{f(\sigma^{\text{g}}_{1:t}, \cdot)} \sum_{i=1}^n \left\{ \sum_{\tau=1}^{T-t} g_i\left(\tau + 1, \sigma^\pi_{1:\tau}, \cdot\right) + \sum_{\tau=T-t+1}^T g_i\left(\tau + 1, \sigma^\pi_{1:\tau}, \cdot\right) \right.$$
$$\left. - \sum_{\tau=t+1}^T g_i\left(\tau + 1, \sigma^{\text{g}+\pi}_{1:\tau}, \cdot\right) - \sum_{\tau=1}^t g_i\left(\tau + 1, \sigma^{\text{g}+\pi}_{1:\tau}, \cdot\right) \right\}$$

$$= 1 + \frac{1}{nT} \frac{1}{f(\sigma^{\text{g}}_{1:t}, \cdot)} \sum_{i=1}^n \left\{ \sum_{\tau=T-t+1}^T g_i\left(\tau + 1, \sigma^\pi_{1:\tau}, \cdot\right) - \sum_{\tau=1}^t g_i\left(\tau + 1, \sigma^{\text{g}+\pi}_{1:\tau}, \cdot\right) \right.$$
$$\left. + \underbrace{\sum_{\tau=1}^{T-t} g_i\left(\tau + 1, \sigma^\pi_{1:\tau}, \cdot\right) - \sum_{\tau=t+1}^T g_i\left(\tau + 1, \sigma^{\text{g}+\pi}_{1:\tau}, \cdot\right)}_{\leq 0} \right\}$$

$$\leq 1 + \frac{1}{nT} \frac{1}{f(\sigma^{\text{g}}_{1:t}, \cdot)} \sum_{i=1}^n \left\{ \sum_{\tau=T-t+1}^T g_i\left(\tau + 1, \sigma^\pi_{1:\tau}, \cdot\right) - \sum_{\tau=1}^t g_i\left(\tau + 1, \sigma^{\text{g}+\pi}_{1:\tau}, \cdot\right) \right\} \qquad (20)$$

Let $\sigma_{1:t}^i := \underbrace{i \oplus i \oplus \cdots \oplus i}_{t \text{ times}}$ denote the sequence of items of length $t$ that consists of all $i$'s. Then, clearly

$$\sum_{\tau=T-t+1}^{T} g_i \left( \tau + 1, \sigma_{1:\tau}^{\pi}, \cdot \right) \leq \sum_{\tau=T-t+1}^{T} g_i \left( \tau + 1, \sigma_{1:\tau}^i, \cdot \right) \tag{21}$$

Combining Eq. (20) with Eq. (21) we get

$$\omega_t \leq 1 + \frac{1}{nT} \frac{1}{f(\sigma_{1:t}^g, \cdot)} \sum_{i=1}^{n} \left\{ \sum_{\tau=T-t+1}^{T} g_i \left( \tau + 1, \sigma_{1:\tau}^{\pi}, \cdot \right) - \sum_{\tau=1}^{t} g_i \left( \tau + 1, \sigma_{1:\tau}^{g+\pi}, \cdot \right) \right\}$$

$$\leq 1 + \frac{1}{nT} \frac{1}{f(\sigma_{1:t}^g, \cdot)} \sum_{i=1}^{n} \left\{ \sum_{\tau=T-t+1}^{T} g_i \left( \tau + 1, \sigma_{1:\tau}^i, \cdot \right) - \sum_{\tau=1}^{t} g_i \left( \tau + 1, \sigma_{1:\tau}^{g+\pi}, \cdot \right) \right\}$$

$$= 1 + \frac{1}{nT} \frac{1}{f(\sigma_{1:t}^g, \cdot)} \sum_{i=1}^{n} \left\{ \sum_{\tau=T-t+1}^{T} g_i \left( \tau + 1, \sigma_{1:\tau}^i, \cdot \right) - \sum_{\tau=1}^{t} g_i \left( \tau + 1, \sigma_{1:\tau}^g, \cdot \right) \right\} \tag{22}$$

*Proof of Theorem 4.* Clearly, both the empirical bounds on $\gamma_t^g$ (Algorithm 2) and $\omega_t^g$ (RHS of Eq. (22)) can be computed in polynomial time. Plugging the values into Theorem 2 and Corollary 2 we get a polynomial time approximation of the empirical bound. □

### E.4 Proof of Theorem 5

In this section, we provide the proof of Theorem 5.

Suppose there are $n$ items, and $T$ is a multiple of $n$. Fix $a$, and assume that $a_i = b_i = a$ and $c_i = 0$ for all $i \in \{1, \ldots, n\}$. We first show a sufficient condition on $a$ under which the greedy policy reduces to the round robin policy.

Recall from Eq. (9) that the recall probability of an item is

$$g_i \left( \tau, \cdot \right) = 2^{\frac{\tau - \ell}{h_i}} \tag{23}$$

where $h_i = 2^{an_i}$ denotes the half life of item $i$, and $n_i$ denotes the number of times item $i$ is presented so far.

Now assume that the greedy algorithm picks item $i$ at $t = 1$. Then, in order for the greedy algorithm not to pick the same item at $t = 2$, we need to make sure that at $t = 2$, the gain of item $i$ is smaller than the gain of the best item. To achieve that, there must exist some other item $j$, such that

$$\Delta \left( j \mid \sigma_1 = i \right) > \Delta \left( i \mid \sigma_1 = i \right)$$

That is,

$$\sum_{t=2}^{T} \left( g_j \left( t, \sigma_1 = i, \sigma_2 = j \right) - g_j \left( t, \sigma_1 = i \right) \right) > \sum_{t=2}^{T} \left( g_i \left( t, \sigma_1 = i, \sigma_2 = i \right) - g_i \left( t, \sigma_1 = i \right) \right)$$

A sufficient condition for the above inequality to hold is

$$g_j \left( T, \sigma_1 = i, \sigma_2 = j \right) - g_j \left( T, \sigma_1 = i \right) = g_j \left( T, \sigma_1 = i, \sigma_2 = j \right)$$
$$> g_i \left( T, \sigma_1 = i, \sigma_2 = i \right) - g_i \left( T, \sigma_1 = i \right)$$

Plugging in the definition of $g_i, g_j$, we get

$$2^{-\frac{T-1}{2a}} > 2^{-\frac{T-1}{2^2 a}} - 2^{-\frac{T}{2a}} \tag{24}$$

It is easy to verify numerically that a sufficient condition for Eq. (24) to hold is

$$a \geq \log T \tag{25}$$

Next, we provide a lower bound on the cost of the round robin algorithm. Let $\sigma_{1:T}$ be the round robin teaching sequence. W.l.o.g., assume that the order of items shown in each round is $1, 2, \ldots, n$. Therefore,

$$
\begin{aligned}
f(\sigma_{1:T}) &= \frac{1}{nT} \sum_{i=1}^{n} \sum_{\tau=1}^{T} g_i\left(\tau + 1, \sigma_{1:\tau}\right) \\
&= \frac{1}{nT} \sum_{i=1}^{n} \sum_{r=1}^{T/n} \sum_{\tau=1}^{n} g_i\left((r-1)n + \tau + 1, \sigma_{1:(r-1)n+\tau}\right) \\
&\geq \frac{1}{nT} \sum_{i=1}^{n} \sum_{r=1}^{T/n} n g_i\left(rn + 1, \sigma_{1:(r-1)n+\tau}\right) \\
&= \frac{1}{T} \sum_{i=1}^{n} \sum_{r=1}^{T/n} g_i\left(rn + i, \sigma_{1:(r-1)n+\tau}\right)
\end{aligned}
$$

For simplicity, define $p_{i,r} = g_i\left(rn + i, \sigma_{1:(r-1)n+\tau}\right)$. We thus have

$$
f(\sigma_{1:T}) = \frac{1}{T} \sum_{i=1}^{n} \sum_{r=1}^{T/n} p_{i,r} \tag{26}
$$

Observe that for $r \in \{1, \ldots, T/n\}$, it holds that

$$
\frac{1 - p_{i,r+1}}{1 - p_{i,r}} \geq \frac{1 - p_{i,r+2}}{1 - p_{i,r+1}}, \text{ and } 1 - p_{i,r} \geq 1 - p_{i,r+1} \tag{27}
$$

From the above inequalities we get

$$
\begin{aligned}
1 - p_{i,r+1} &= (1 - p_{i,r}) \frac{1 - p_{i,r+1}}{1 - p_{i,r}} \\
&\leq (1 - p_{i,r}) \frac{1 - p_{i,r}}{1 - p_{i,r-1}} \\
&\leq (1 - p_{i,r-1}) \frac{1 - p_{i,r-1}}{1 - p_{i,r-2}} \cdot \frac{1 - p_{i,r}}{1 - p_{i,r-1}} \\
&\leq (1 - p_{i,r-1}) \left(\frac{1 - p_{i,r-1}}{1 - p_{i,r-2}}\right)^2 \\
&\leq (1 - p_{i,1}) \left(\frac{1 - p_{i,2}}{1 - p_{i,1}}\right)^r
\end{aligned}
$$

Therefore, we have

$$
\begin{aligned}
\sum_{r=1}^{T/n} (1 - p_{i,r}) &\leq (1 - p_{i,1}) + (1 - p_{i,1}) \frac{1 - p_{i,2}}{1 - p_{i,1}} + \cdots + (1 - p_{i,1}) \left(\frac{1 - p_{i,2}}{1 - p_{i,1}}\right)^{T/n - 1} \\
&= \sum_{r=1}^{T/n} (1 - p_{i,1}) \left(\frac{1 - p_{i,2}}{1 - p_{i,1}}\right)^{r-1} \\
&= \frac{(1 - p_{i,1}) \left(1 - \left(\frac{1 - p_{i,2}}{1 - p_{i,1}}\right)^{T/n}\right)}{1 - \left(\frac{1 - p_{i,2}}{1 - p_{i,1}}\right)} \\
&\leq \frac{(1 - p_{i,1})^2}{p_{i,2} - p_{i,1}} \tag{28}
\end{aligned}
$$

Combining Eq. (26) with Eq. (28) we get

$$
f(\sigma_{1:T}) = \frac{1}{T} \sum_{i=1}^{n} \sum_{r=1}^{T/n} p_{i,r}
$$

$$= 1 - \frac{1}{T} \sum_{i=1}^{n} \sum_{r=1}^{T/n} (1 - p_{i,r})$$

$$\geq 1 - \frac{1}{T} \sum_{i=1}^{n} \frac{(1 - p_{i,1})^2}{p_{i,2} - p_{i,1}}$$

$$\overset{(a)}{=} 1 - \frac{n}{T} \frac{(1 - p_{i,1})^2}{p_{i,2} - p_{i,1}}$$

where step (a) is due to the fact that $p_{i,1} = 2^{-n/2^a}$, and $p_{i,2} = 2^{-n/2^{2a}}$ for all $i$.

Now suppose that we would like to lower bound the utility $f(\sigma_{1:T})$ by $1 - \epsilon$. Therefore,

$$\frac{n}{T} \frac{(1 - p_{i,1})^2}{p_{i,2} - p_{i,1}} \leq \epsilon \tag{29}$$

While it is challenging to solve Eq. (29) in an analytical form, we consider a stronger condition to simplify the calculation. Consider a configuration of $a$ which also satisfies the following inequality

$$1 - p_{i,2} \leq \frac{1 - p_{i,1}}{2} \tag{30}$$

Therefore, a sufficient condition for Inequality (29) to hold is

$$\frac{(1 - p_{i,1})^2}{p_{i,2} - p_{i,1}} = \frac{(1 - p_{i,1})^2}{(1 - p_{i,1}) - (1 - p_{i,2})} \overset{\text{Eq. (30)}}{\leq} \frac{(1 - p_{i,1})^2}{(1 - p_{i,1}) - \frac{1 - p_{i,1}}{2}} = 2(1 - p_{i,1}) \leq \frac{\epsilon T}{n}$$

Plugging in $p_{i,1} = 2^{-n/2^a}$ into the above inequality, we get

$$2^{-n/2^a} \geq 1 - \frac{\epsilon T}{2n} \tag{31}$$

Now, let us consider the following two cases:

C1 $1 - \frac{\epsilon T}{2n} > 0$ (that is, $\epsilon < 2n/T$). In this case, we get

$$a \geq \log \left( \frac{n}{\log \left( \frac{1}{1 - \epsilon T/(2n)} \right)} \right)$$

$$= \log n - \log \log \left( \frac{1}{1 - \epsilon T/(2n)} \right)$$

$$\overset{(a)}{\geq} \log n - \log \left( \left( \frac{1}{1 - \epsilon T/(2n)} \right) - 1 \right)$$

$$= \log \left( \left( \frac{2n^2}{\epsilon T} \right) - n \right)$$

where step (a) is by the inequality $\log(x) \leq x - 1$ for $x > 0$. A feasible configuration of $a$ satisfying the above inequality is

$$a \geq \log \left( \frac{2n^2}{\epsilon T} \right) \tag{32}$$

It is easy to verify that Condition Eq. (32) also satisfies our additional constraint Eq. (30).

C2 A second case is $\epsilon \geq 2n/T$. In this case, Eq. (31) holds for all $a$, and we only need to find a feasible configuration of $a$ that satisfies Eq. (30). A suitable choice of such a constraint is

$$a \geq \log (3n) \tag{33}$$

Combining Eq. (25) Eq. (32) and Eq. (33) we obtain

$$a \geq \max \left\{ \log T, \log (3n), \log \left( \frac{2n^2}{\epsilon T} \right) \right\}$$

which finishes the proof.