[Reviews · NeurIPS 2019]

Reviewer 1



I think this is a very solid but unspectacular paper - explained below. Looks like a safe paper to accept as a poster. 1. The paper studies an interesting problem of next item selection to maximize retention in memory-related tasks (German, Biodiversity). The difference between this work and prior works is multi-concept; while this is an important contribution, it is not a fundamental one. 2. I like the theoretical contributions: the formulation is solid, the results obtained using submodularity are more or less expected. The model-specific result for exponential forgetting curves is somewhat useful - memory strength and the number of concepts have a log relationship, which makes sense. 3. For the human user experiments, the results are pretty good - statistically significant improvement despite using only 80/320 users. However, these results are heavily restricted; the whole session lasts only 25 minutes, which makes it hard to conclude that the same results will hold for real-life learning settings. I also wish there is a parameter sensitivity study (lines 264-270), since the authors simply ran the algorithm with some prior parameter choices and didn't tune them. Can you use the collected data to fit some parameters and compare them with your choices? Writing is good. I read the author response and it is good - I'll keep my score.

Reviewer 2



Originality: The work is a nice combination of developments in a few fields -- applying analysis and concepts from machine teaching to concept learning in forgetful humans. To the best of my knowledge, the theoretical analysis is novel. However, they missed some critical work in the field and missed an opportunity to compare their method to very relevant pre-existing work. This includes the following references: 1) Patil, Zhu, Kopec, & Love (2011). Optimal teaching for Limited-Capacity Human Learners. NIPS. (http://papers.nips.cc/paper/5541-optimal-teaching-for-limited-capacity-human-learners.pdf) -- they examine concept learning and machine teaching, but with a different variant on limiting capacity of the ability "to use" exemplars. It is similar and may even be isomorphic to the case in this submission (I doubt it's isomorphic). 2) Nosofsky, R. N., Sanders, C. A., Zhu, X., & McDaniel, M. A. (2019). Model-based search for optimal natural-science-category training exemplars: A work in progress. Psychonomic Bulletin & Review, 26, 48-76. Quality: The strongest portion of the article is the derivation of theoretical results. Although the simulations and behavioral experiments are a valuable contribution, the contribution is weakened due to the weak baselines chosen by the authors. Ignoring missing previous work, I would have liked to have seen a comparison to optimal teaching to a "forgetless" learner or how different levels of forgetting affected teaching and learning. There are also the models mentioned before which would have served as good baselines. Here are a few other minor issues: Line 190-192: HLR memory model. This really has a much longer provenance prior to Settles and Meeder (2016). It would be good to acknowledge previous researchers (references through the Pavlik and Anderson 2005 on ACT-R activation functions and learning should get you there. References in Walsh, Gluck, Gunzelmann, Jastrezembski, Krusmark, Myung, Pitt, & Zhang, 2018. Mechanisms underlying the Spacing Effect in Learning: A Comparison of Three Computational Models. Journal of Experimental Psychology: General could also help) who have used forgetting functions of a similar form. Line 282-284: The statistical tests aren't appropriate for binary outcome responses. The authors instead should use logistic regression or chi-squared tests based on contingency tables. Clarity: The paper is well-written, though I did have some difficulty following their mathematical derivation at times. One reason might be their use of \gamma, which is more traditionally a decay rate of some sort. I also was unclear on the precise definition of \tau as it seemed to change from section 3.2 to 3.3. I appreciate the lack of space and partially my expertise being more in the computational cognitive science than machine teaching analysis. Author feedback response: Thank you for your thoughts on my criticism of the submission. I apologize for my confusion and I appreciate your clarification. It may be worth mentioning that in the revised manuscript as other readers might have it. I am glad you are including the hypothesis testing statistics in the revised manuscript.

Reviewer 3



Originality: I think the specific teaching model proposed in the paper has never been considered in the literature. Quality: Owing to time constraints, I only managed to check the proofs of Theorems 1 and 2; I think they are correct. Overall, I think the quality of the main paper is generally very good, with very few typos. Clarity: The paper is quite clearly written. Significance: The development of a framework for modelling the teaching of multiple concepts to memory-limited learners is quite significant. Minor Comments: - Page 2, lines 67-68: Perhaps state what the acronym ACT-R stands for (just like other model acronyms used in the same sentence). - Page 4, equation (5): Should this definition/notation be extended to the conditional marginal gain of teaching a _sequence_ of concepts at time t? (On page 15, in the last equality before line 476, \Delta is applied to such a sequence.) - Page 6, Section 4.2: Perhaps explain what a HLR memory model is in more detail. - Page 9, references [8] and [22]: I suggest either spelling out the acronym PNAS in [8] or using the acronym PNAS in [22] (i.e., stick to only one formatting style). - Page 10, reference [27]: "The Generalization of [`student's'] problem..." - Page 14, line 449: Do we need to use the submodularity of \mu to show that g_i(\tau+1, (\sigma_{1:min(\tau,t)},\cdot)) \leq 1$ (if so, it might be helpful to mention this, since string submodular functions were not defined in the paper)? - Page 14, line 460: "...denote such [a] case by..." - Page 15, definition of conditional marginal gain of a policy: How is item i (mentioned in Line 468) used in the definition? It might be helpful to give an intuitive explanation for why, on the right-hand side of (12), one takes the concatenation of \sigma_{1:t} with \sigma^{\pi}(\Phi) (similarly for y_{1:t} and y^{\pi}(\Phi)); in particular, why does the sequence of items selected by \pi from t' = 1 to t' = t appear twice (once in \sigma_{1:t} and again in \sigma^{\pi}(\Phi))? (Did I interpret the definition wrongly?) - Page 16, inequality between lines 481 and 482: I could not see why this inequality follows directly from Definition 2. According to Definition 2, \omega_t is computed by taking the maximum over all (\sigma_{1:t},y_{1:t}) of the maximum expectation over all policies \pi with respect to (\sigma^\pi(\Phi),y^{\pi}(\Phi)); however, imposing the additional condition \Phi \sim (\sigma_{1:t},y_{1:t}) seems to reduce the value of E[f(\sigma_{1:t} \oplus \sigma^{\pi}(\Phi),y_{1:t}\oplus y^{\pi}(\Phi)) - f(\sigma^{\pi}(\Phi),y^{\pi}(\Phi))]/f(\sigma_{1:t},y_{1:t}). - Page 16, inequality (17): Do we need to take the expectation of the second summand? * Response to author feedback: Thank you very much for the detailed feedback. I am keen on following future work on this topic, especially any possible long-term experiments on language learning using the paper's algorithm. I will keep my current score.

[Author Response · NeurIPS 2019]

We thank the reviewers for their valuable suggestions. Please find our answers (**A**) for each reviewer (**R**) below.

**R1, R3**: *Parameter sensitivity study (ln 264–270)*

**A**: We had conducted a sensitivity study on simulated learners before choosing the HLR parameters for our user study. These detailed results were omitted from the original submission. We report them below and we will include these results in the revision. In this experiment, we consider two groups of concepts: "easy/common" concepts with $\theta = (10, 5, 0)$, and "hard/rare" concepts with $\theta = (3, 1.5, 0)$. Other configurations are kept the same as our user study, with $T = 40$, $n = 15$ and $s = 10$. We vary the number of "easy" concepts from $\{0, 1, \ldots, 8\}$ (i.e., up to 50% of the concepts being easy), and consider four types of teachers: (i) "easy": $\theta = (10, 5, 0)$ for all concepts; (ii) "hard": $\theta = (3, 1.5, 0)$ for all concepts; (iii) "true": using the true parameters for each concept; (iv) "robust": $\theta = (6, 2, 0)$ for all concepts. We plot the performances of these different teachers measured by the two metrics considered in simulations (i.e., the objective value and future recall). As shown in the figures, the "robust" teacher performs well on both metrics, and hence is used for our user study on the German dataset.

**R1, R3**: *Time scale of real-life learning settings*
**A**: After the publication of this work, it is quite conceivable to apply these ideas in real-life language learning scenarios. We consider collaborating with existing language learning platforms as a natural step for future work.

**R1**: *Fit the parameters of the HLR model, and compare them with the current parameters of choice*
**A**: Thanks for the suggestion. Indeed, one can infer $\theta$ for each concept from historical data. We have collected $800$ user entries from the random teacher on the German dataset (and $3200$ entries on Biodiversity), and it is possible to take existing user study histories and fit an HLR model to get an estimate of $\theta$. We plan to include the results in the revision.

**R2**: *Relevant pre-existing work: optimal teaching with exemplars; references on HLR memory model*
**A**: Thanks for pointing us to these references. We will certainly include them in the revision. However, there might be some misunderstanding about the differences between the exemplar-based setting (Patil et al. 2011, Nosofsky et al. 2018) and the setting of our work. Patil et al. (2011) and Nosofsky et al. (2018) investigated the problem of choosing the optimal exemplars (based on the Generalized Context Model) for teaching a *classification* task; whereas for our case, the exemplars for each class are already given (in other words, we have only one "exemplar" per class), and we aim at optimally teaching the learner to *memorize* the (label of) exemplars. It is unclear how one can adapt the algorithm to our setting, as they are addressing two orthogonal problems. We will explain these points in the updated paper.

**R2**: *Alternative baselines: (1) Optimal "forgetless" leaner; (2) different levels of forgetting*
**A**: Under our problem setting (as explained in our previous response), if the learner is "forgetless", then after teaching each concept the recall probability becomes 1, leading to a trivial teaching scenario (by showing each concept once). To see the effect of different levels of forgetting, please refer to our sensitivity study results in response to **R1**.

**R2**: *Significance tests*
**A**: We performed $\chi^2$ tests (with contingency tables where rows are algorithms and columns are observed outcomes) and obtained very similar statistics with Table 1 (see below). We will include these statistics in the updated paper.

| | German | | | | Biodiversity | | | | Biodiversity (common) | | | | Biodiversity (rare) | | | |
|---|---|---|---|---|---|---|---|---|---|---|---|---|---|---|---|---|
| | GR | LR | RR | RD | GR | LR | RR | RD | GR | LR | RR | RD | GR | LR | RR | RD |
| *p*-value ($\chi^2$ tests) | – | 0.0652 | 0.0197 | 0.0151 | – | 0.0017 | <0.0001 | <0.0001 | – | 0.3111 | 0.8478 | 0.0047 | – | 0.0001 | <0.0001 | <0.0001 |

**R3**: *Upper-bound on $F(\pi^g)$*
**A**: This is a very interesting suggestion. It will be interesting to establish an upper bound for the greedy or optimal algorithm under particular model configurations, e.g., to provide a necessary condition for achieving a certain target utility under the HLR model (similar to Thm 5). We will further explore this question as future work.

**R3**: *Teaching interface: Special consideration for teaching "two consecutive time steps"*
**A**: In our teaching interface, there was no gap between two teaching iterations. The learner could copy the answer from the previous iteration to the next question if the same concept was shown. Therefore, we treat this case specially to mitigate such effect. An alternative way is to introduce a small break between two teaching iterations.

**R3**: *Minor Comments*
**A**: Thanks for the detailed suggestions. We will fix all the issues with careful proofreading, especially in the Appendix. A few specific answers: (i) *Proof of Thm 1 (Page 14, line 449)*: Yes, we will clarify that $g_i(\cdot) \leq 1$ is due to $\mu$ being submodular, and (ii) *Proof of Thm 2 (Page 16)*: Inequality (17) does not require an expectation of the second summand as $(\sigma_{1:t}^g, y_{1:t}^g)$ is the *observed* history. For the inequality between lines 481-482: this is a good point – we will revise this inequality by imposing the condition only on the first part of the expectation. This does not affect the rest of the proof.

[Meta-Review · NeurIPS 2019]

This paper proposes a framework for modelling the process of teaching multiple concepts to a learner (e.g. a human student) with limited memory. The authors provide empirical justification for their approach and certain theoretical guarantees. Interestingly, there is the possibility that the insights from this paper could be applied to language learning in humans, which would I think represent an intriguing and uncommon domain of application for a Neurips paper. There is, of course, also potential application to human (or machine) teaching of machines. Significantly, the two reviewers expressed not only their approval of the work, but also the fact that they learned something of value from reading it. I share these sentiments. Another factor in my recommendation is how satisfied the reviewers were with the author response. Although scores did not change, the solid response of the authors seems to have been much appreciated. For these reasons, and the fact that the work tackles and interesting a non-typical learning situation, I think the paper should be accepted and will make a welcome addition to the conference programme.